# Novel Biosynthesis of Graphene-Supported Zero-Valent Iron Nanohybrid for Efficient Decolorization of Acid and Basic Dyes

Mahmoud Samy [1], Marwa Elkady [2,3,*], Ayman Kamal [4,5], Noha Elessawy [6], Sahar Zaki [4]
and Marwa Eltarahony [4,*]

1 Department of Public Works Engineering, Faculty of Engineering, Mansoura University,
Mansoura 35516, Egypt
2 Chemical and Petrochemical Engineering Department, Egypt-Japan University of Science and Technology,
New Borg El-Arab City, Alexandria 21934, Egypt
3 Fabrication Technology Department, Advanced Technology and New Materials Research Institute (ATNMRI),
City of Scientific Research and Technological Applications, New Borg El-Arab City, Alexandria 21934, Egypt
4 Environmental Biotechnology Department, Genetic Engineering and Biotechnology Research
Institute (GEBRI), City of Scientific Research and Technological Applications (SRTA-City),
Alexandria 21934, Egypt
5 Faculty of Agriculture, Alexandria University, Alexandria 21544, Egypt
6 Computer Based Engineering Applications Department, Informatics Research Institute IRI,
City of Scientific Research & Technological Applications (SRTA-City), Alexandria 21934, Egypt
* Correspondence: marwa.elkady@ejust.edu.eg (M.E.); m_eltarahony@yahoo.com (M.E.);
Tel.: +20-100-978-2536 (Marwa Eltarahony)

**Abstract:** Herein, respiratory nitrate reductases (NAR) were utilized in the biosynthesis of zero-valent iron (ZVI) graphene nanocomposite as a simultaneous reducing and capping agent, for the first time, to efficiently adsorb methylene blue (MB) and direct red-81 (DR-81). Under anaerobic conditions, the greenly synthesized graphene was incubated with iron precursor in the presence of crude-NAR enzyme for 48 h to obtain the ZVI graphene composite followed by characterizing this composite using physiochemical analyses. Scanning and transmission electron microscopy, energy dispersive X-ray spectroscopy and X-ray diffraction techniques assured the chemical composition and the interaction between ZVI and graphene. The influences of operating conditions such as contact time, pH and adsorbent dose on the adsorption efficacy were explored in the case of ZVI graphene, graphene and ZVI. ZVI graphene nanocomposite displayed the highest removal efficiency of MB and DR-81 compared to graphene and ZVI-NPs. The removal percentages of DR-81 and MB by ZVI graphene nanocomposites were $88.3 \pm 2.66\%$ and $87.6 \pm 2.1\%$, respectively, at pH 7, adsorbent dose 20 mg/50 mL, initial MB or DR-81 concentration of 10 mg/L and shaking speed of 150 rpm. A pseudo first-order model could describe the adsorption kinetics, and the adsorption mechanism was discussed. The promising results of the current study support the potential of the recruitment of ZVI graphene nanocomposites in eliminating various pollutants from industrial effluents on a larger scale. Further, the prepared nanohybrid can be used in other applications such as photocatalysis, Fenton and persulfate activation processes.

**Keywords:** decolorization; direct red-81; graphene; methylene blue; nitrate reductases; operating parameters; zero-valent iron

## 1. Introduction

The expansion of industrial activities such as cosmetics, leather, ink, textiles and printing contributes to the release of heavily contaminated effluents containing dyes to water streams [1,2]. Azo dyes constitute almost 50% of the global production (700,000 ton/year) and during the dyeing process, about 20% of the dyes can be discharged to water sources [3]. The presence of dyes in water bodies inhibits light penetration and photosynthesis and reduces the dissolved oxygen ratios resulting in threats and risks to humans and animals [4].

Therefore, it is imperative to effectively treat effluents containing dyes before discharge to the environment. Many physical, chemical and biological treatment methods have been proposed for the removal of dyes [5–7]. The azo dyes are highly resistant to being degraded by microorganisms and they are toxic to bacteria which make the biological treatment ineffective [8]. Similarly, the removal of dyes by physical–chemical treatment processes such as adsorption, photocatalysis, ozonation and membrane filtration has been investigated [9–12]. However, these methods are associated with some drawbacks such as high costs and the production of toxic by-products [13].

The adsorption process is a promising, inexpensive, simple and efficient treatment process for the removal of industrial effluents (e.g., dyes) compared to previously mentioned techniques [14]. Activated carbon is a commonly employed adsorbent for the removal of bio-resistant pollutants (dyes) from aqueous solutions [15]. However, activated carbon challenges the high cost and requirement of frequent regeneration which obstruct the reusability and full-scale application [16]. Hence, there is an increasing need for the preparation of effective adsorbents with low costs. Zero-valent iron (ZVI) has recently pulled wide heed due to its low cost, high reactivity, green impact, high surface area as well as its capability to reduce oxidized contaminants [12,17]. ZVI is commonly prepared via the liquid-phase reduction in iron salts using strong reducing agents (e.g., sodium borohydride). In spite of the simplicity, short time and high reactivity of the liquid reduction method, this method suffers from shortcomings such as high costs, toxicity of the reducing agent and poor stability of prepared nanoparticles [18]. Moreover, the release of nanomaterials prepared from harmful chemicals after the treatment process has a negative impact on the aquatic and terrestrial environments [19]. Therefore, the efforts have been directed to prepare nanoparticles using ecofriendly materials (e.g., bacteria, plants, algae). In this study, we employed respiratory nitrate reductase (NAR) enzyme as a green reducing and stabilizing agent to replace toxic reducing agents needed in the preparation of ZVI nanoparticles. Due to the toxicity of metal ions to bacteria, bacteria start to protect themselves and then, metal ions can bioaccumulate or biomineralize into the cells after the reduction of metal ions to a lower state [20]. The biosynthesis of ZVI can reduce the cost, protect the environment from toxic chemicals and produce stable nanoparticles.

Another problem facing ZVI is the fast aggregation and formation of the iron oxide layer which reduces the reactivity and reducibility of ZVI [12,21]. To overcome this problem, pure ZVI can be supported with different materials such as silica, kaolinite, starch and graphene [22–24]. Several studies in recent years recorded the efficiency of graphene and its oxide, in particular as a nanosorbent, in different remediation applications of polluted environments by the dint of its high theoretical surface area (~2620 $m^2/g$), high electrical conductivity and stability [5]. In this work, a ZVI graphene nanocomposite was prepared by reducing iron salt in the presence of graphene, which was produced from plastic waste, by the catalysis of NAR enzyme. The use of graphene can reduce the agglomeration and the oxidation of ZVI, facilitate the electron transfer from ZVI surface to the pollutant and improve the dispersibility of the nanoparticles [25]. Moreover, the conversion of plastic waste (polyethylene terephthalic) to graphene participates in the management of environmental problems related to the release of plastic wastes to the environment and reduction in costs related to the disposal of plastic waste [26]. Liu et al. (2014) prepared ZVI supported by graphene for the removal of phosphorus from aqueous solutions [25]. They stated that the prepared composite attained a high removal of phosphorus.

In this study, the novel procedures for the biological synthesis of ZVI graphene nanocomposite were provided. The synthesized ZVI graphene nanocomposite was utilized for the removal of methylene blue (MB) and direct red-81 (DR-81). The effects of operating parameters such as contact time, pH and adsorbent dosage were investigated. The removal mechanism and adsorption kinetics of MB and DR-81 were explored.

## 2. Materials and Methods

### 2.1. Materials

Plastic bottle wastes were collected from the garbage of City of Scientific Research and Technological Applications in Borg Al Arab, Alexandria. Sodium citrate ($Na_3C_6H_5O_7$, 98%), hydrochloric acid (HCl, 97%), sodium chloride (NaCl, 97%), magnesium sulfate heptahydrate ($MgSO_4 \cdot 7H_2O$, 99%), ferrous chloride tetrahydrate ($FeCl_2 \cdot 4H_2O$, 95%), sodium molybdate ($Na_2MoO_4 \cdot 2H_2O$, 99%), copper sulfate ($CuSO_4 \cdot 5H_2O$, 95%), cobalt chloride hexahydrate ($CoCl_2 \cdot 6H_2O$, 98%), manganese chloride tetrahydrate ($MnCl_2 \cdot 4H_2O$, 97%), zinc sulfate ($ZnSO_4$, 93%), potassium phosphate ($KH_2PO_4$, 95%), dipotassium hydrogen phosphate ($K_2HPO_4 \cdot 7H_2O$, 97%) and potassium nitrate ($KNO_3$, 98%) were procured from Sigma-Aldrich and used without modifications.

### 2.2. Biosynthesis of ZVI Graphene Nanocomposite

Initially, graphene and ZVI-NPs were prepared following the procedures described in detail by El Essawy et al. (2017) and Zaki et al. (2019) [27,28]. For the preparation of ZVI graphene nanocomposite, respiratory membrane-bound nitrate reductase enzyme (NAR) was recruited as reducing and stabilizing agent simultaneously in an in situ synthesis process. NAR enzyme as well as electron shuttling molecules commence nitrate reduction and shuttle electrons to iron ions. Then, the oxidation states of metal ions can change via redox reaction until reaching zero-valent state and forming ZVI-NPs [28,29]. In the presence of graphene, the enzymatically generated ZVI-NPs bind to graphene particles, forming ZVI graphene nanocomposites. Firstly, about $10^8$ CFU/mL (0.5 McFarland) of bionanofactory strain *P. mirabilis* 10B was precultured anaerobically in the optimized broth that enhances and induces the growth of NAR enzyme, of the following ingredients (g/L): sodium citrate 7.5, NaCl 0.5, $MgSO_4 \cdot 7H_2O$ 0.12, $FeCl_2 \cdot 4H_2O$ 0.24, $Na_2MoO_4 \cdot 2H_2O$ 0.015, $CuSO_4 \cdot 5H_2O$ 0.06, $CoCl_2 \cdot 6H_2O$ 0.04, $MnCl_2 \cdot 4H_2O$ 0.03, $ZnSO_4$ 0.31, $KH_2PO_4$ 3.0, $K_2HPO_4 \cdot 7H_2O$ 1.0 and $KNO_3$ 5.0 at pH 7.0. The inoculated broth was incubated anaerobically at 30 °C for 48 h in anaerobic jar. After incubation, a considerable bacterial biomass was harvested by centrifugation at 10,000 rpm for 20 min. The harvested cells were washed several times by sterile distilled water and the, suspended in 80 mM potassium phosphate buffer (pH 7.0) and exposed to mild osmotically shock disruption followed by centrifugation at 5000 rpm for 5 min to eliminate unbroken cells and cell debris. Thereafter, the obtained spheroplast was subjected to centrifugation at 10,000 rpm, 4 °C for 60 min. The precipitated pellets were resuspended in 80 mM potassium phosphate buffer (pH 7.0) and utilized as a complete membrane-bound $\alpha\beta\gamma$ complex of NAR. Finally, 0.2 gm of graphene was added to crude-NAR suspension that was supplemented by 1.5 mM of Fe $(NO_3)_3 \cdot 9H_2O$. The overall mixture was incubated under anaerobic conditions for 48 h at 30 °C. The obtained ZVI graphene nanocomposite was collected, washed and dried at 80 °C for 2 h for subsequent characterization and application.

### 2.3. Characterization of ZVI Graphene Nanocomposite

The crystallographic information of ZVI graphene nanocomposite was investigated by an X-ray diffraction (XRD, Shimadzu-7000, USA) using Cu K$\alpha$ radiation ($\lambda$ = 1.54056 Å), voltage of 40 KV, current of 30 mA and scan speed of 10 °/min. The incorporation of ZVI in carbon matrix of graphene was affirmed via the specification of elemental composition using energy dispersive X-ray spectroscopy (EDX) analyzer and elemental mapping combined with transmission electron microscope (TEM, JEOL JSM 6360LA, Japan). Additionally, TEM and scanning electron microscopy (SEM) (JEOL JEM-1230) with an accelerating voltage of 200 kV were employed to study the morphology of the synthesized composite. Raman spectra (Jaso, Japan) were recorded to specify the molecular structure of the synthesized nanoparticles. Raman spectra were set in the 4000–400 $cm^{-1}$ region with a step size of 1 $cm^{-1}$. Fourier transform infrared spectroscopy (Shimadzu, FTIR-8400S) was performed to specify the functional groups using KBr pellets in the 4000–400 $cm^{-1}$ region with a step size of 1 $cm^{-1}$. Surface area was estimated using Belsorp-max automated apparatus (BEL Japan)

and the sample was degassed at 200 °C for 3 h. The data of nitrogen (adsorption–desorption isotherms) were recorded at 77.53 K.

*2.4. Experimental Procedures*

A stock solution with a concentration of 1000 mg/L was prepared for direct red-81 (DR-81) and methylene blue (MB) by dissolving 1 g of the dye in 1 L of distilled water. The adsorption batch experiments were conducted in 100-mililiter beaker with 50 mL of the dye solution and the solution was shaken at a speed of 150 rpm. The effects of different parameters such as contact time (15–120) min, adsorbent dose (20, 40, 60, 100 mg/50 mL), and pH (3, 7, 9, 11) were studied at an initial dye concentration of 10 mg/L. Samples were withdrawn and then centrifuged prior to the measurements by UV–Visible spectrophotometer (Labomed model, Inc., Los Angeles, CA, USA) at wavelengths of 500 and 664 nm for DR-81 and MB, respectively. The comparison between the performance of pure ZVI-NPs, graphene and ZVI graphene nanocomposite was performed at pH 7, initial dye concentration of 10 mg/L, adsorbent dose of 20 mg/50 mL and shaking speed of 150 rpm. The adsorption kinetics were investigated using pseudo first- and second-order models and intraparticle diffusion on a time range (0–90 min) at pH 7, adsorbent dose 20 mg/50 mL, initial MB or DR-81 concentration of 10 mg/L and shaking speed of 150 rpm. The equations and discussion of these models were provided in detail in our previous work [9]. In reusability study, the particles were compiled after each run and left at room temperature to dry, thence they can be used in next cycles.

## 3. Results and Discussion

### 3.1. Characterization of ZVI Graphene Nanocomposite

In the present study, the NAR enzyme was successfully utilized as a catalyzing and stabilizing agent to synthesize ZVI graphene nanohybrid. Generally, the biological approaches of nanomaterial synthesis are advantageous over other traditional physicochemical means. They are characterized by their cost-effectiveness, non-toxicity, biosafety and ecofriendliness. They reckon on the utilization of biological molecules in the reduction of bulk parent materials into their nanoform; neglecting by such way the usage of hazardous reductants, high-temperature or energy-intensive processing [28,29]. It is noteworthy to mention the characteristic role of nitrate reductases (NRs), which belong to oxidoreductases groups. They are mainly catalyzing organic material oxidation and play crucial function in pollution remediation and eventually participate in the nitrogen cycle. Recently, NRs were reported in various investigations addressing nanostructure synthesis in green biological methods [29,30]. The ZVI graphene nanocomposite was analyzed by various characterization techniques. SEM and TEM images of pure graphene were provided in the Supplementary Materials (Figure S1) to show the changes in the morphology in the case of the composite. Figure 1a,b shows TEM and SEM images of ZVI graphene biocomposites with particle size ranged from 2.3 to 7 nm, indicating the incorporation of ZVI-NPs in the graphene matrix. However, the magnetic properties of ZVI-NPs resulted in the aggregation of ZVI-NPs in chain-like forms.

Additionally, the EDX pattern in Figure 1c confirmed the presence of both carbon and iron at their characteristic positions (0.277 and 6.4 keV, respectively) with atomic percentages of 50.8% and 36.5%, respectively. Other peaks at 0.39 and 2.3 keV were ascribed to nitrogen and sulfur with atomic ratios of 9.2 and 3.5%, respectively. Interestingly, the existence of nitrogen and sulfur might be owing to the bounded bacterial biomolecules during the synthesis process, such as amino groups ($-NH_3^+$) and sulfhydryl (SH) groups, which could eventually provide the nanocomposite with self-functionalizing and stabilizing properties. However, a Cu peak was also detected due to the copper grid that was used to hold the samples. The XRD pattern in Figure 1d demonstrates the XRD pattern of bare graphene showing only the diffraction planes of graphene. The XRD pattern in Figure 1e affirmed the interaction between the graphene and ZVI-NPs. The peak at 24.67° is attributed to the (002) diffraction plane of graphene [3]. The sharp peak at 44.4° is attributed to the

(110) diffraction plane of ZVI-NPs, whereas the peaks at 31.7°, 35.56°, 51.82° and 59.92° are attributed to the iron oxides formed on the ZVI surface due to the oxidation that might be occurred during washing, drying and processing of the nanocomposite [2,31].

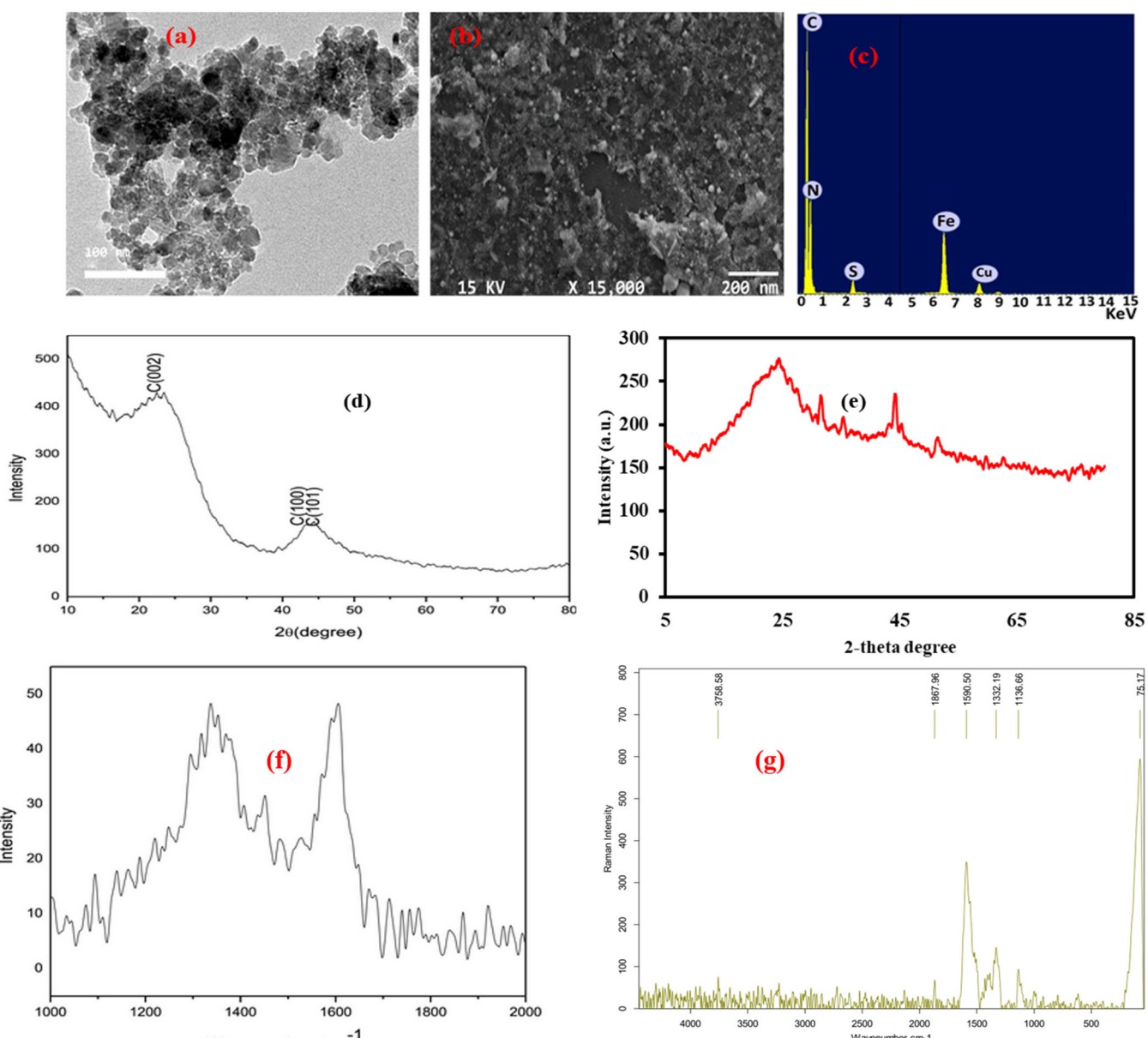

**Figure 1.** Physicochemical properties (**a**) TEM image, (**b**) SEM, (**c**) EDX pattern of ZVI graphene, (**d**) XRD pattern of graphene, (**e**) XRD of ZVI graphene, (**f**) Raman spectra of graphene and (**g**) Raman spectra of ZVI graphene [27].

Raman spectra of pristine graphene are given in Figure 1f, showing only the D and G bands of graphene. Figure 1g shows the Raman spectra of ZVI graphene. The peaks at 1332.19 cm$^{-1}$ and 1590.5 cm$^{-1}$ are ascribed to the D and G bands of graphene [25]. The peaks at wavenumbers lower than 1000 cm$^{-1}$ (around 75 cm$^{-1}$, 550 cm$^{-1}$ and 750 cm$^{-1}$) are attributed to the Fe-O bond, confirming the presence of iron oxides on the surface of ZVI-NPs [31]. Supplementary Materials Figure S2a,b shows the adsorption–desorption isotherms and FTIR spectra of graphene. The surface area was 721.7 m$^2$/g. The bands in FTIR spectra at 3447, 1636, 1219 and 1105 cm$^{-1}$ are indexed to O-H, C=O, O-H, C-O-C and C-O bonds, respectively [27]. Additionally, the band at 1600 cm$^{-1}$ is imputed to the

C==C bond [27]. Figure S2c,d demonstrates the FTIR spectra and adsorption–desorption isotherms of ZVI graphene. The presence of the hydroxyl group (O-H) was confirmed through the bands at 1610 and 3420 cm$^{-1}$. The band at 1350 cm$^{-1}$ is assigned to the COO$^-$ group [32]. Moreover, the band at 1020 cm$^{-1}$ is attributed to the C-O bond. The bands at 476 and 628 cm$^{-1}$ are imputed to the Fe-O bond. The surface area of ZVI graphene was estimated (55 m$^2$/g). Figure S2e,f,g shows the elemental mapping of graphene and ZVI graphene [33]. In the case of graphene, only carbon was detected. Whereas carbon, iron and oxygen were detected in the elemental mapping of the composite.

### 3.2. Application of ZVI Graphene Nanocomposite in Cationic and Anionic Dye Removal

As observed in Figure 2a,b, the alterations in the UV–vis spectra (from 200 to 800 nm) of DR-81 and MB were monitored upon treatment with ZVI graphene nanocomposite, ZVI and graphene. The spectrum patterns of DR-81 and MB (controls) exhibited the main absorption peaks at 500 and 664 nm for DR-81 and MB, respectively, due to the azo bond, which almost disappeared in the supernatant of the treated samples especially in the case of ZVI graphene. Such vanishing of both peaks reflected the successful removal of both dyes. Our results are coincident with those investigated by Samy et al. (2020) [34]. However, the adsorption process was confirmed by SEM in Figure 2c,d. The pores in the case of ZVI graphene nanocomposite approximately disappeared due to the filling of pores by the dye molecules confirming the high adsorption performance of the dyes by the biosynthesized composite. These things considered, new peaks were formed due to the formation of chemical bonds between the iron oxide layer and dye molecules, as revealed by Raman spectroscopy (Figure 2e,f).

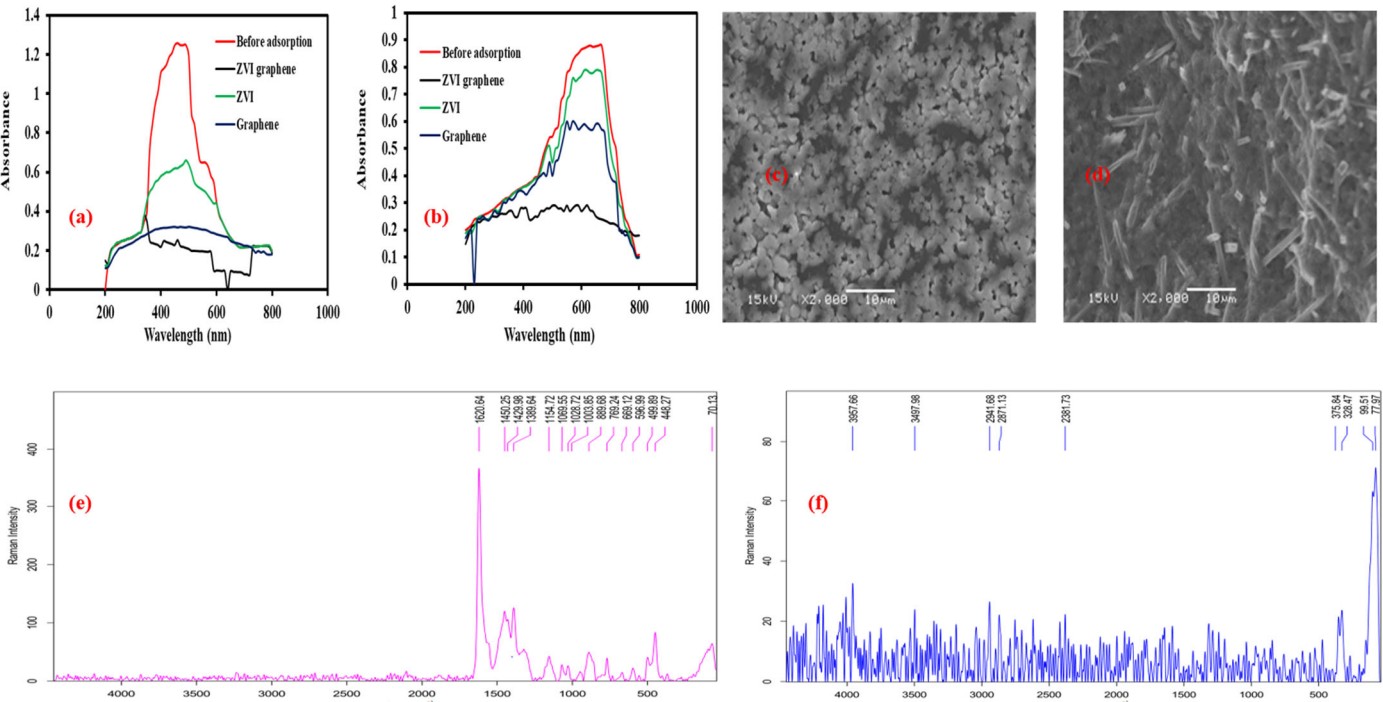

**Figure 2.** Absorption spectra of (**a**) DR-81 and (**b**) MB before and after adsorption over ZVI graphene, ZVI and graphene; SEM image of ZVI graphene after adsorption of (**c**) DR-81 and (**d**) MB; Raman spectra of ZVI graphene after adsorption of (**e**) DR-81 and (**f**) MB.

Additionally, XRD and FTIR of ZVI graphene after adsorption of DR-81 and MB are given in Supplementary Materials Figure S3a,b. The intensity and transmittance decreased after the adsorption of DR-81 and MB. The change in the intensity and transmittance might be due to the bonds formed between the adsorbent's surface and adsorbate. EDX of ZVI graphene after adsorption in Figure S3c,d showed the introduction of Cl and Na in

the case of MB and DR-81, reaffirming the bonding between the adsorbent's surface and the adsorbate.

### 3.3. Effect of Contact Time

A comparison between the adsorption performance of ZVI-NPs, graphene and ZVI graphene nanocomposite was conducted to specify the required time for achieving equilibrium and choose the best material, as shown in Figure 3a,b. The experiments were conducted at pH 7, adsorbent dose 20 mg/50 mL, initial MB or DR-81 concentration of 10 mg/L and shaking speed of 150 rpm. The results showed that 90 min was the equilibrium time in the cases of MB and DR-81. The prolonged equilibrium time indicated the porous structure and high surface area of the prepared composite. Gajera et al. (2022) reported an equilibrium time of 2 h [35]. Moreover, the ZVI graphene nanocomposite showed higher adsorption performance towards MB and DR-81 compared to ZVI-NPs and graphene. The removal efficiencies of DR-81 were $61.4 \pm 1.6\%$, $49.8 \pm 1.7\%$ and $88.3 \pm 2.66\%$ in the cases of graphene, ZVI-NPs and ZVI graphene nanocomposite, respectively, whereas the removal efficiencies of MB were $68.8 \pm 3.1\%$, $53.3 \pm 2.39\%$ and $87.6 \pm 2.1\%$ after 90 min with the same order. The extension of the reaction time did not greatly improve the removal percentage because the pores became saturated and the adsorbents' surfaces reached an equilibrium state [25]. The biogenic ZVI graphene nanocomposite attained a removal efficiency of 91.8% and 91.3% for DR-81 and MB, respectively, after 120 min. The synergistic effects of graphene and ZVI-NPs were the major reason for such high removal ratios. The graphene participated in the enhancement of ZVI-NPs dispersion and reduction in aggregation [36]. Graphene exhibited higher performance than ZVI-NPs due to its high surface area, while pure ZVI-NPs might agglomerate which decreased the surface area, thereby reducing the adsorption efficiency.

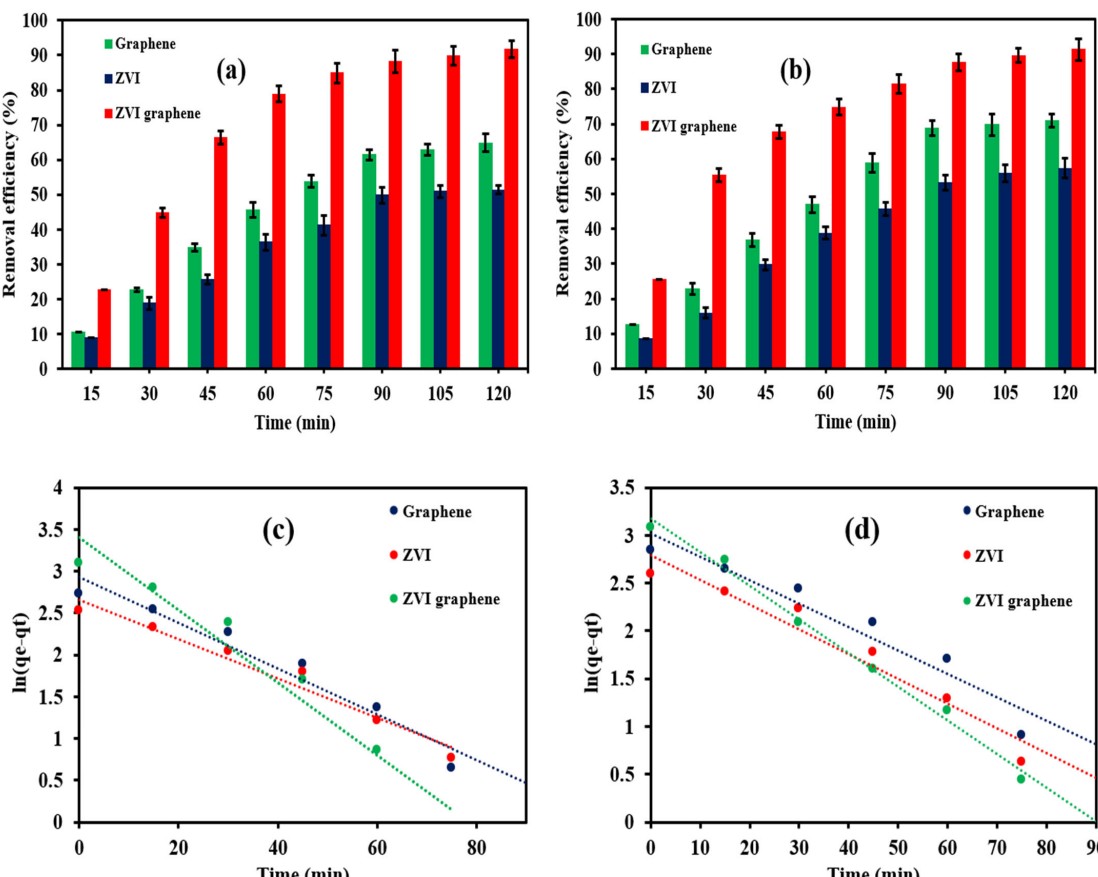

**Figure 3.** Effect of contact time on the adsorption efficiency of (**a**) DR-81 and (**b**) MB, and pseudo first-order kinetic model in the case of (**c**) DR-81 and (**d**) MB.

The rate constants of the pseudo first-order model were estimated in the case of graphene, ZVI and ZVI graphene via the linear plot of $\ln(q_e-q_t)$ versus time, as shown in Figure 3c,d. The high coefficient of determination values and the slight difference between experimental adsorption capacity and estimated adsorption capacity from the first-order model at equilibrium affirmed the suitability of the first-order model to describe adsorption kinetics. The rate constants were 0.0273, 0.0236 and 0.0434 $min^{-1}$ for graphene, ZVI and ZVI graphene, respectively, in the case of DR-81, whereas the rate constants were 0.026, 0.0245 and 0.0352 $min^{-1}$ for the synthesized materials with the same order in the case of MB. The high-rate constant in the case of the composite compared to graphene and ZVI was in agreement with the results obtained from the comparison between the three materials. Second-order and intraparticle diffusion models were also provided in the Supplementary Materials (Figure S4). Table S1 shows the constant rate ($K_1$ and $K_2$), coefficient of determination ($R^2$) and adsorption capacity at equilibrium ($q_e$) in the cases of first- and second-order models as well as the constants and $R^2$ of the intraparticle diffusion model. According to $R^2$ values in Table S1, the second-order model could not describe the adsorption process. Additionally, obtained $q_e$ from the second-order model is far from the experimental value. The intraparticle diffusion model displayed the multilinearity of the adsorption process.

### 3.4. Effect of Adsorbent Dose

Figure 4a,b shows the effect of adsorbent dose on the removal efficiency of DR-81 and MB at pH 7, shaking speed at 150 rpm and initial dye concentration of 10 mg/L after a contact time of 90 min. The increase in the graphene dose from 20 mg/50 mL to 60 mg/50 mL resulted in the raising of the removal efficiency from 61.4 ± 1.6% to 88.9 ± 1.9% in the case of DR-81 and from 68.8 ± 3.1% to 89.8 ± 2.4% in the case of MB. However, the increase in graphene dose above 60 mg/50 mL did not show any amelioration in the removal efficiency in both dyes. Regarding ZVI, the increase in the ZVI dose above 60 mg/50 mL showed slight improvement in the adsorption efficiency. In the case of ZVI graphene, upon increasing the dose from 20 mg/50 mL to 40 mg/50 mL, the enhancement of the removal efficiency from 88.3 ± 2.66% to 96.88 ± 1.9% in the case of DR-81 and from 87.6 ± 2.1% to 98.66 ± 1.9% in the case of MB was noticed. The raising of ZVI graphene to 100 mg/50 mL just increased the removal efficiency to 99.77 ± 3.1% and 99.6 ± 2.6% in the cases of DR-81 and MB, respectively. In general, the raising of adsorbent dose increases the number of binding sites which contributes to the amelioration of adsorption performance. Howbeit, doses above the optimum value might decrease or cause a slight improvement of the removal efficiency because of the possible agglomeration of nanoparticles. The agglomeration of nanoparticles could decrease the surface area and, consequently, reduce the active adsorption sites [37]. Mensah et al. (2022) reported the same observation and showed that the increase in the adsorbent dose above 0.2 mg/mL did not boost the adsorption ratio [3].

The stability of the prepared composite was examined under five repetitive runs (450 min, 90 min for each run) for the adsorption of DR-81 and MB, as shown in Supplementary Materials Figure S5. The slight decrease in adsorption percentage in the sequential runs assured the stability of the composite. Howbeit, the decrease in the adsorption ratio might be due to the loss of adsorbent during sampling.

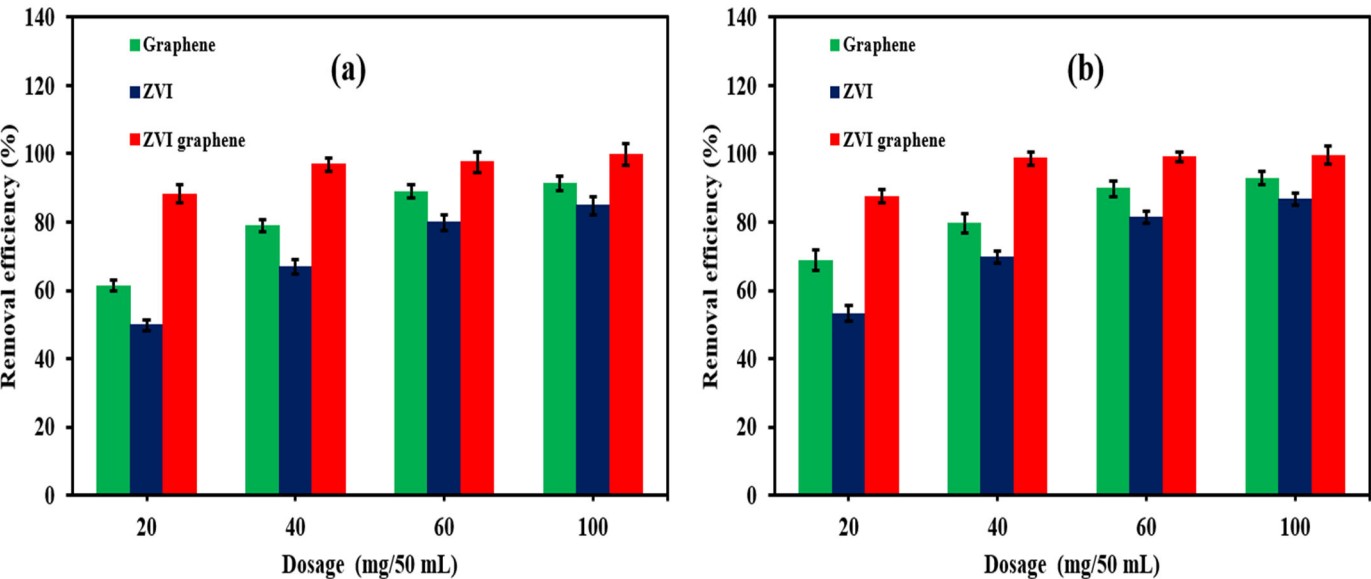

**Figure 4.** Effect of adsorbent dose on the removal efficiency of (**a**) DR-81 and (**b**) MB.

### 3.5. Effect of pH

The effect of pH was studied by varying pH from 3 to 11 (using 1 M of HCl or NaOH) at initial dye concentration of 10 mg/L, adsorbent dose of 40 mg/50 mL and shaking speed of 150 rpm, as shown in Figure 5. Point of zero charge (PZC) of ZVI is nearly eight, as reported by Sun et al. (2015) [38]. At pH < PZC, the adsorbent's surface charge becomes positive, and DR-81 is an ionic dye. Therefore, the removal efficiency was improved in acidic and neutral conditions due to the attractive forces between the adsorbent's surface and DR-81 molecules. The elimination efficiency increased from $96.88 \pm 1.67\%$ to $99.4 \pm 1.9\%$ by decreasing pH from 7 to 3 using the ZVI graphene nanocomposite. On the other hand, the removal efficacy went down in alkaline conditions due to the repulsive forces between negatively charged adsorbent's surface and anionic dye molecules. The removal efficiencies were $85.6 \pm 2.3\%$ and $78.9 \pm 2.67\%$ at pH 9 and 11, respectively, using the ZVI graphene nanocomposite. In the case of MB, the removal efficiencies decreased at pH 3 and it was $81.4 \pm 1.9\%$ due to the repulsive forces between positively charged sorbent's surface and cationic dye. However, at neutral pH, the removal efficiency was high. The increase in pH from 3 to 7 resulted in the raising of hydroxyl ions which could be adsorbed on the adsorbent's surface and, consequently, MB as a cationic dye could be adsorbed on the surface [39]. At high pH values, the removal efficiencies decreased to $88.7 \pm 1.4\%$ and $80.1 \pm 1.1\%$ at pH 9 and 11, respectively, in spite of the attractive forces between the adsorbent and adsorbate. At alkaline conditions, ZVI could be easily corroded and, consequently, iron hydroxides would be formed and precipitated [39]. The formed iron hydroxides could block the active sites and inhibit the electron transfer between ZVI and target pollutants which resulted in the decrease in removal efficiency [40]. Hamdy et al. (2018) reported the decrease in MB removal efficiency in alkaline condition due to the block of active sites of ZVI by the corrosion products [41]. In the case of the adsorption of MB and DR-81 over ZVI at pH > 7, the adsorption efficiency decreased due to the accelerated corrosion of ZVI, resulting in the formation of iron hydroxides that could block the binding sites and suppress the electron transfer [40]. At pH ≤ 7, the adsorption ratio went up in the case of DR-81 due to the attraction between DR-81 molecules and ZVI surface, whereas the adsorption percentage declined in the case of MB due to the repulsive forces [39]. The adsorption at pH 3 was lower than that of pH 7 due to the competition between MB molecules and Cl⁻ on ZVI surface [42]. At pH ≤ 7, the graphene surface charge is positive, whereas it was negative at pH > 7, as reported by Yang et al. (2022) [43]. So, the adsorption efficiency improved in the case of DR-81 at pH < 7 due to the attraction forces, while the adsorption ratio of DR-81 went down at pH > 7 due to the repulsive forces. In the case of MB, the adsorption

proportion decreased at pH < 7 due to the repulsive forces and the adsorption went up at pH > 7 owing to the attractive forces. In the case of MB at pH 7, the adsorbed hydroxyl ions on the adsorbent's surface might increase the attractive forces, thus improving the degradation ratio.

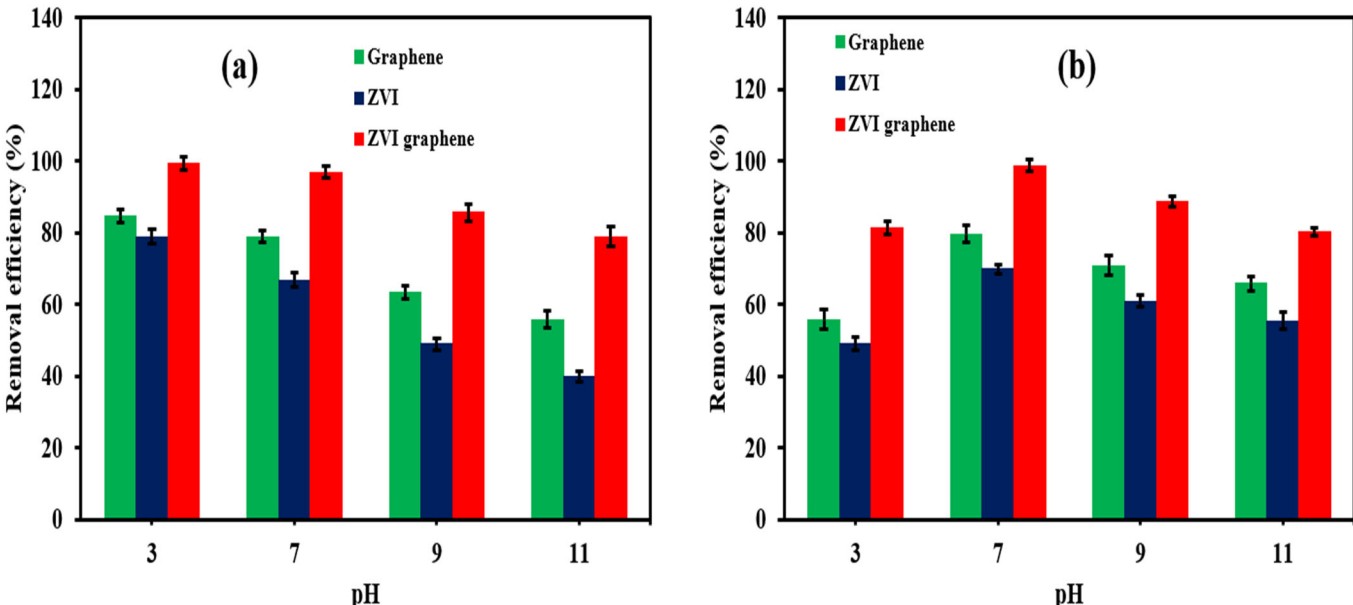

**Figure 5.** Effect of pH on the removal efficiency of (**a**) DR and (**b**) MB.

### 3.6. Adsorption Mechanism

Figure 6 demonstrates the adsorption mechanism of DR-81 and MB on ZVI graphene nanocomposite's surface. The adsorption of MB on the adsorbent's surface might be owing to the hydrogen bonding between OH and COOH groups of the adsorbent and NH group of MB [44]. In the case of DR-81, the adsorption took place via hydrogen bonding between $SO_3^-$, NH and N=N of DR-81 and the hydroxyl group in the adsorbent [45]. Additionally, the hydrogen bonding could happen between COOH of the adsorbent and the NH group of DR-81. The adsorption could take place due to the electrostatic interaction between the $COO^-$ group (deprotonated form of COOH) of the adsorbent and $NH^+$ of MB. Moreover, the adsorption might occur due to the electrostatic interaction between $Na^+$ and the deprotonated form of COOH ($COO^-$) in the case of DR-81 [46]. Further, the adsorption of MB or DR-81 might take place due to the π-π stacking between aromatic groups of MB or DR-81 and oxygen of the hydroxyl group in the prepared adsorbent [45,47]. Additionally, the adsorption could take place due to the Yoshida hydrogen bonding between benzene ring of DR-81 or MB and H in the hydroxyl group [45,48]. The decolorization of DR-81 and MB might also take place due to the reduction by atomic hydrogen (H*) formed owing to the reaction between electrons and $H^+$ on the adsorbent's surface [49]. Furthermore, the formed iron hydroxides could adsorb DR-81 and MB. The adsorption of DR-81 or MB on ZVI graphene is governed by hydrogen bonding, electrostatic interaction (attractive and repulsive forces), Yoshida hydrogen bonding and π-π stacking. So, the increase or the decrease in the adsorption percentage can be affected by the attractive or repulsive forces between the adsorbent and adsorbate, as explained in Section 3.5. The adsorption in the case of graphene could take place via the same mechanisms (hydrogen bonding, electrostatic interaction (attractive and repulsive forces), Yoshida hydrogen bonding and π-π stacking based on its functional groups, as previously described in the literature) [3]. In the case of ZVI, adsorption could occur via hydrogen bonding, electrostatic interaction (attractive and repulsive forces), Yoshida hydrogen bonding and π-π stacking [49]. Additionally, the removal of dyes via ZVI might take place by reduction and precipitation [50].

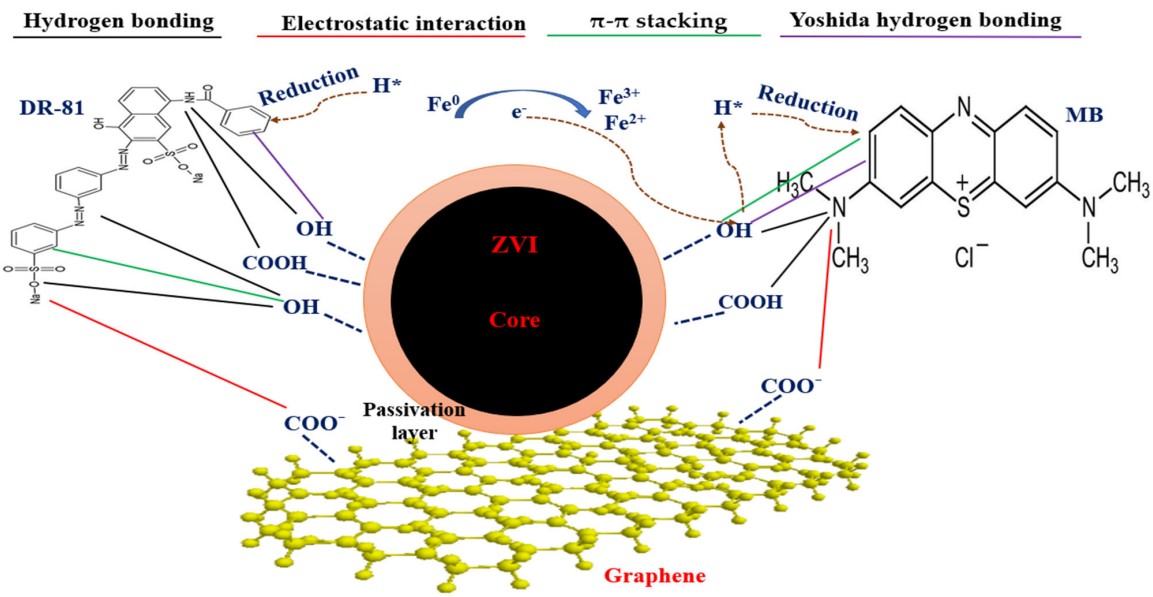

**Figure 6.** Adsorption mechanism of DR-81 and MB over ZVI graphene nanocomposite.

## 4. Conclusions

The excellent interaction between the graphene prepared from plastic waste and the biosynthesized ZVI under the catalysis of bacterial NAR enzyme was confirmed by different analytical techniques such as TEM, SEM, EDX, XRD and Raman spectra. The ZVI graphene nanocomposite achieved the highest removal efficiency compared to ZVI-NPs and graphene. The ZVI graphene nanocomposite attained a removal efficiency of 88.3 ± 2.66% and 87.6 ± 2.1% in the case of DR-81 and MB, respectively, after an equilibrium contact time of 90 min. The increase in adsorbent dose from 20 to 40 mg/50 mL resulted in the increase in the removal efficiency from 88.3 ± 2.66% to 96.88 ± 1.9% in the case of DR-81, compared to 87.6 ± 2.1% and 98.66 ± 1.9% in the case of MB. High removal efficiency of DR-81 was attained at acidic and neutral conditions, whereas the removal performance of MB was improved at neutral conditions using the ZVI graphene nanocomposite. The ZVI graphene had the highest adsorption rate in the case of MB and DR-81. The adsorption of MB or DR-81 on ZVI graphene surface could be due to hydrogen bonding, electrostatic interaction, Yoshida hydrogen bonding and π-π stacking. Such biogenic nanocomposite can contribute to lowering the toxicity and the costs of chemicals employed in the synthesis of ZVI-NPs and can overcome the hazards related to the plastic wastes. Additionally, the utilization of plastic wastes in the synthesis of graphene can contribute to the efficient management, reduction in greenhouse gases resulted from the burning or/and landfilling of plastic waste and scalable production of graphene. The high performance, biocompatibility and inexpensiveness of ZVI graphene nanocomposites can encourage decision makers to the full-scale treatment of industrial effluents by the biologically fabricated ZVI graphene nanocomposite. Further, we plan and advise other researchers to employ this efficient nanohybrid in other applications such as photocatalysis, Fenton reaction or persulfate activation system expecting high performance.

**Supplementary Materials:** The following supporting information can be downloaded at: https://www.mdpi.com/article/10.3390/su142114188/s1.

**Author Contributions:** Writing—original draft, M.S., M.E. (Marwa Elkady) and M.E. (Marwa Eltarahony); validation, M.S., S.Z. and A.K.; investigation, M.S., N.E. and M.E. (Marwa Eltarahony); formal analysis, M.S., N.E. and M.E. (Marwa Elkady); writing—review and editing, M.S., M.E. (Marwa Elkady) and M.E. (Marwa Eltarahony); methodology, M.E. (Marwa Eltarahony) and A.K.; supervision, M.E. (Marwa Eltarahony), S.Z. and M.E. (Marwa Elkady). All authors have read and agreed to the published version of the manuscript.

**Funding:** This research received no external funding.

**Institutional Review Board Statement:** Not applicable.

**Informed Consent Statement:** Not applicable.

**Data Availability Statement:** All data are provided.

**Acknowledgments:** The authors are grateful to Mansoura University for publication fee support.

**Conflicts of Interest:** The authors declare no conflict of interest.

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
