# Peer review of "Novel Biosynthesis of Graphene-Supported Zero-Valent Iron Nanohybrid for Efficient Decolorization of Acid and Basic Dyes"

_sustainability, doi:10.3390/su142114188_

Round 1
Reviewer 1 Report
Recently, the applications of graphene oxide and its derivatives in the dye adsorption (removal) process have been widely investigated by scientists. In this work, the nanocomposite of iron and graphene synthesized by the green synthesis process was used for decolorization in dye adsorption performance. I suppose that the novelty in this research is highly great and capable of deeply investigating to expanse the applied fields for this material and similar ones. However, the manuscript needs to be made a lot of revisions:

Author Response
Dear
Editor-in-Chief
Sustainability - MDPI
We are pleased to send our responseto the comments of reviewers, included at the bottom of this letter.All added text and modifications are highlighted in the revised manuscript. We have answered point-by-point response to these comments as follows,
_____________________________________________________________
Reviewer #1:
Comment:
Recently, the applications of graphene oxide and its derivatives in the dye adsorption (removal) process have been widely investigated by scientists. In this work, the nanocomposite of iron and graphene synthesized by the green synthesis process was used for decolorization in dye adsorption performance. I suppose that the novelty in this research is highly great and capable of deeply investigating to expanse the applied fields for this material and similar ones. However, the manuscript needs to be made a lot of revisions:
Thank you for praising of our manuscript. We responded to your valuable comments as given below.
- 1. The spelling and grammatical structures in the manuscript should be improved.
Answer:
Thank you for your valuable comment. We made a rigorous proofreading to correct all grammatical errors. Besides, the final version was revised by a native English speaker.
2.Based on my experiences, I suggest that the content of the abstract might be focused on the mainideas and shorter than it used to be. In brief, the abstract follows the format: (1) the purposes of the manuscript,(2) the main method used in this research, and (3) the principle results, major conclusion, and orient applied
development. Thus, I suggest the authors make a revision to the abstract.
Answer:
Thank you for your valuable comment.The abstract was revised to be more precise and to follow the mentioned order.
The abstract was revised as follows:
“Herein, respiratory nitrate reductases (NAR) were utilized in the biosynthesis of zero-valent iron (ZVI)@graphene nanocomposite as a simultaneous reducing and capping agent, for the first time to efficiently adsorb methylene blue (MB) and direct red-81 (DR-81). Under anaerobic conditions, the greenly synthesized graphene was incubated with iron precursor in the presence of crude-NAR enzyme for 48 hours to obtain ZVI@graphene composite followed by characterizing this composite using physiochemical analyses. Scanning and transmission electron microscopy, energy dispersive X-ray spectroscopy and X-ray diffraction techniques assured the chemical composition and the interaction between ZVI and graphene. The influences of operating conditions such as contact time, pH and adsorbent dose on the adsorption efficacy were explored in the case of ZVI@graphene, graphene and ZVI. ZVI@graphene nanocomposite displayed the highest removal efficiency of MB and DR-81 compared to graphene and ZVI-NPs. The removal percentages of DR-81 and MB by ZVI@graphene nanocomposite were 88.3±2.66% and 87.6±2.1%, respectively at pH 7, adsorbent dose 20 mg/50 mL, initial MB or DR-81 concentration of 10 mg/L and shaking speed of 150 rpm. Pseudo first-order model could describe the adsorption kinetics and the adsorption mechanism was discussed. The promising results of the current study support the potential of the recruitment ofZVI@graphene nanocomposite in eliminating various pollutants from industrial effluents on a larger scale. Further, the prepared nanohybrid can be used in other applications such as photocatalysis, Fenton and persulfate activation processes.”
- The format of the manuscript should be entirely at once. All paragraphs need to be formatted asjustified alignment.
Answer:
Thank you for your valuable comment.All paragraphs were formatted as justified alignment.
- One of my biggest concerns about the study: is a lack of statistical analysis needs attention. It isnecessary to add the statistical analysis for all results.
Answer:
Thank you for your valuable comment. The error bars were added to all figures after performing each experiment two times. The removal percentages were given as “Removal percentage (%)±standard deviation”.
- It is essential to provide a graphical abstract to summarize the whole research.
Answer:
Thank you for your valuable comment. A graphical abstract was provided to describe the whole research.
- Please give the formation mechanism of the ZVI@graphene nanocomposite using NAR enzymes.
Answer:
Thank you for your valuable comment.The formation mechanism of ZVI@graphene nanocomposite using NAR enzymes was provided.
The following mechanism was added:
“NAR enzyme as well as electron shuttling molecules commence nitrate reduction and shuttle electrons to iron ions. Then, the oxidation states of metal ions can change via redox reaction till reaching zero-valent state and forming ZVI-NPs [28, 29]. In the presence of graphene, the enzymatically generated ZVI-NPs bind to graphene particles forming ZVI@graphene nanocomposites.”
- The TEM and SEM images of graphene precursors are required to be provided as controlswith ZVI@graphene nanocomposite and given in the supplementary material file.
Answer:
Thank you for your valuable comment.Actually, TEM and SEM images of graphene were presented in our previous study and added to the supplementary file.
- Also, the XRD and Raman patterns of graphene are added in the same diagrams ofthe ZVI@graphene nanocomposite for comparison.
Answer:
Thank you for your valuable comment.XRD and Raman patterns of graphene were addedfrom our previous workin the same diagram with ZVI@graphene.
- The FITR spectroscopy and element mapping of graphene and the ZVI@graphenenanocomposite are carried out to determine the composition and the elemental distribution in the samples,especially iron.
Answer:
Thank you for your valuable comment. FTIR and elemental mapping of graphene (from our previous work) and ZVI@graphenewere added to the supplementary file.
- I refer to the specific surface area (BET) calculation of the ZVI@graphene nanocomposite that mustbe added.
Answer:
Thank you for your valuable comment. Adsorption-desorption isotherms were provided in the supplementary file and surface area value was added.
- I worry about that how can the authors demonstrate the stability of the ZVI@graphenenanocomposite.
Answer:
Thank you for your valuable comment. We preformed repetitive runs on the nanocomposite to check its stability over prolonged operating time as shown in Figure A5. The adsorption efficiency was high in five runs assured the high stability of the composite. Howbeit, the slight decrease in following runs might be due to the adsorbent’s particles leaving the adsorption process during sampling.
- The authors must be provided in detail the kinetics and equilibrium of the other adsorption modelsinstead of the pseudo-first-order model.
Answer:
Thank you for your valuable comment. The kinetic models were provided in the supplementary file. Moreover, Table S1 was added in the supplementary to show constant rates, R2 and qe of kinetic models.
- After the adsorption process, did the desorption process be conducted?
Answer:
Thank you for your valuable comment. In this work, we focused on investigating the adsorption performance of ZVI@graphene. Moreover, we want to simulate the real application, so reusability experiments were performed without regeneration (desorption). The reusability test implied high adsorption efficiency in five cycles in the case of DR-81 and MB confirming the high stability and the potential for large-scale application for the treatment of real wastewater.
- The authors need to provide SEM and FESEM images, FTIR spectra, XRD patterns, Raman spectra,and element mapping of the ZVI@graphene nanocomposite after both adsorption and desorption processes with
respect to DR-81 and MB. The explanation of these results is also analyzed as required.
Answer:
Thank you for your valuable comment. Raman spectra and SEM analyses of ZVI@graphene after adsorption of DR-81 and MB were addedconfirming the adsorption of MB and DR-81 on ZVI@graphene surface as shown in Figure (2).Additionally, we added XRD, EDX and FTIR of ZVI@graphene after adsorption in the supplementary file (Figure A3). EDX can show the chemical composition after adsorption the same as elemental mapping analysis. Raman analysis is now in the maintenance stage, and we think XRD, EDX and FTIR are enough to confirm the adsorption of DR-81 and MB on ZVI@surface and show the changes that take place on the adsorbent’s surface. FESEM analysis is not available in our institution and performing this analysis in an external lab will take long time. The above-mentioned analyses were not performed for the composite after desorption, because we did not perform desorption process as explained in comment 13.
- I suggest that the authors should be constructed the structure and rearranged the results, if possible,for logic and rationality throughout the manuscript. It can be available as a supplementary material file inaddition to containing the supporting results.
Answer:
Thank you for your valuable comment. The results were rearranged as well as many modifications were performed to make the manuscriptmore logic.Moreover, supplementary materials were added to the supplementary file to support the results.
- I recommend the authors refer to the following studies for a clearer and more detailed explanation ofthe DR-81 and MB adsorption mechanism of the ZVI@graphene nanocomposite. Individuals the adsorptionmechanism of graphene, zero-valent iron is required. Then, the generally simultaneous mechanism of both
components should be proposed as the provable feasibility of themechanism: https://doi.org/10.1016/j.surfin.2021.101023,https://doi.org/10.3390/ijerph17165817, https://doi.org/10.3390/nano10050917, https://doi.org/10.1016/j.enmm.2021.100584.
Answer:
Thank you for your valuable comment. The adsorption mechanism was improvedaccording to the proposed studies.
The following discussion was added:
“Figure 6 demonstrates the adsorption mechanism of DR-81 and MB on ZVI@graphene nanocomposite’s surface. The adsorption of MB on the adsorbent’s surface might be owing to the hydrogen bonding between OH and COOH groups of the adsorbent and NH group of MB [44]. In the case of DR-81, the adsorption took place via hydrogen bonding between SO3-, NH and N=N of DR-81 and hydroxyl group in the adsorbent [45]. Additionally, the hydrogen bonding could happen between COOH of the adsorbent and NH group of DR-81. The adsorption could take place due to the electrostatic interaction between COO- group (Deprotonated form of COOH) of the adsorbent and NH+ of MB. Moreover, the adsorption might occur due to the electrostatic interaction between Na+ and deprotonated form of COOH (COO-) in the case of DR-81 [46]. Further, the adsorption of MB or DR-81 might take place due to the π-π stacking between aromatic groups of MB or DR-81 and oxygen of hydroxyl group in the prepared adsorbent [45,47]. Additionally, the adsorption could take place due to the Yoshida hydrogen bonding between benzene ring of DR-81 or MB and H in hydroxyl group [45,48]. The decolorization of DR-81 and MB might also take place due to the reduction by atomic hydrogen (H*) formed owing to the reaction between electrons and H+ on the adsorbent’s surface [49]. Furthermore, the formed iron hydroxides could adsorb DR-81 and MB. The adsorption of DR-81 or MB on ZVI@grapheneis governed by hydrogen bonding, electrostatic interaction (attractive and repulsive forces), Yoshida hydrogen bonding and π-π stacking. So, the increase or the decrease of adsorption percentage can be affected by the attractive or repulsive forces between the adsorbent and adsorbate as explained in Section 3.5. The adsorption in the case of graphene could take place via the same mechanisms (hydrogen bonding, electrostatic interaction (attractive and repulsive forces), Yoshida hydrogen bonding and π-π stacking based on its functional groups as previously described in the literature [3].In the case of ZVI, adsorption could occur via hydrogen bonding, electrostatic interaction (attractive and repulsive forces), Yoshida hydrogen bonding and π-π stacking [49]. Additionally, the removal of dyes via ZVI might take place by reduction and precipitation [50].”
Figure 6. Adsorption mechanism of DR-81 and MB over ZVI@graphene nanocomposite.

Reviewer 2 Report
Manuscript Number: Sustainability-1908385-v1
Title: “NAR enzyme mediates zero-valent iron@graphene nanocomposite: novel synthesis, characterization and efficient decoloriza- 3 tion of Acid and Basic Dyes” by Mahmoud Samy et al.
In this paper, authors are reporting the research work conducted on zero-valent iron@graphene nanocompo- 2 site. The composite has been synthesized and characterized using various significant methods of experimental research in the field of nano-composites. The authors have carried out well-designed experiments and achieved the defined objectives with good satisfactory results.
The following comments should be addressed by the authors before the manuscript could be accepted for publication.
1. In the title “iron@graphene nanocomposite” is used, is it possible to reframe (improve) the title with more appropriate and more attractive specific wordings?
2. In the abstract “48 h” is this the standard way of using unit hours? Instead, you can write 48 hours.
3. The keywords are generally written; reframe the keywords with more appropriate words from the current work.
4. In section 2.3 the characterization method should be more clearly presented.
5. Before results and discussion why section 2.4 is included?
6. The English has to be corrected, for example in section 3.1 “In the present study, the nanocomposite was synthesized successfully by the aid of NAR enzyme as a catalysing and stabilizing agent in this bottom-up approach.” What does this means? I think the sentences have to be reframed.
7. In figure 1, the sub-figures are not clearly generated, redraw the figures with high resolution for clarity.
8. Same heading for section 2.4 and 3.2. It is really confusing.
9. In section 3.5 Figure 5. There is no comparative inference on the effect of pH for three different sample studies. Improve the result analysis part.
10. Sections 3.4 and 3.6 are they come under the same methodology? Any differences between the two, and the discussion on results is not enough in both sections. These sections, 3.4-3.6 can be improved.
11. The adsorption mechanism can be improved with the significant results obtained.
12. All the sections need improvement with a clearer explanation.
13. Improve the resolution of the figures.
14. The conclusion section can also be improved by using more significant findings of the study with futuristic applications of the study.
With all the above clear comments I recommend a thorough revision of the manuscript.
Author Response
Dear
Editor-in-Chief
Sustainability - MDPI
We are pleased to send our responseto the comments of reviewers, included at the bottom of this letter.All added text and modifications are highlighted in the revised manuscript. We have answered point-by-point response to these comments as follows,
_____________________________________________________________
Reviewer #2:
Comment:
In this paper, authors are reporting the research work conducted on zero-valent iron@graphenenanocompo- 2 site. The composite has been synthesized and characterized using various significant methods of experimental research in the field of nano-composites. The authors have carried out well-designed experiments and achieved the defined objectives with good satisfactory results.
Answer:
Thank you for praising of our manuscript. We responded to your valuable comments as given below.
- In the title “iron@graphene nanocomposite” is used, is it possible to reframe (improve) the title with more appropriate and more attractive specific wordings?
Answer:
Thank you for your valuable comment.The title was reframed to be more appropriate and attractive.
The title was reframed to be:
“Novel biosynthesis of graphene-supported zero-valentironnanohybrid for efficient decolorization of acid and basic dyes”
- In the abstract “48 h” is this the standard way of using unit hours? Instead, you can write 48 hours.
Answer:
Thank you for your valuable comment. The abbreviation of hours is frequently written as “h”. Howbeit, to avoid misunderstanding, we replaced “h” in the abstract with “hours”.
- The keywords are generally written; reframe the keywords with more appropriate words from the current work.
Answer:
Thank you for your valuable comments. The keywords were reframed to be more specific and appropriate.
The new keywords can be as follows:
decolorization; direct red-81; graphene; methylene blue; nitrate reductases; operating parameters; zero-valent iron
4.In section 2.3 the characterization method should be more clearly presented.
Answer:
Thank you for your valuable comment.The section 2.3 “Characterization of ZVI@graphene nanocomposite” was revised to be clearer.
The following discussion was added:
“The crystallographic information of ZVI@graphene nanocomposite was investigated by an X-ray diffraction (XRD, Shimadzu-7000, USA) using Cu Kα radiation (λ = 1.54056 Å), voltage of 40 KV, current of 30 mA and scan speed of 10o/min. The incorporation of ZVI in carbon matrix of graphene was affirmed via the specification of elemental composition using energy dispersive X-ray spectroscopy (EDX) analyzer and elemental mapping combined with transmission electron microscope (TEM, JEOL JSM 6360LA, Japan). Additionally, TEM and scanning electron microscopy (SEM) (JEOL JEM-1230) with an accelerating voltage of 200 kV were employed to study the morphology of the synthesized composite. Raman spectra (Jaso, Japan) were recorded to specify the molecular structure of the synthesized nanoparticles. Raman spectra were set in the 4000-400 cm-1 region with a step size of 1 cm-1. Fourier transform infrared spectroscopy (Shimadzu, FTIR-8400S) was performed to specify the functional groups using KBr pellets in the 4000-400 cm-1 region with a step size of 1 cm-1. Surface area was estimated using Belsorp-max automated apparatus (BEL Japan) and the sample was degassed at 200 °C for 3 h. The data of nitrogen (adsorption-desorption isotherms) were recorded at 77.53 K.”
5.Before results and discussion why section 2.4 is included?
Answer:
Thank you for your comment. Section 2.4 was added to show the details of experimental work including the preparation of synthetic dye solution and the details of adsorption set-up.
- The English has to be corrected, for example in section 3.1 “In the present study, the nanocomposite was synthesized successfully by the aid of NAR enzyme as a catalysing and stabilizing agent in this bottom-up approach.” What does this means? I think the sentences have to be reframed.
Answer:
Thank you for your comment. We made a rigorous proofreading to correct all grammatical errors. Besides, the final version was revised by a native English speaker.The sentence in section 3.1 “In the present study, the nanocomposite was synthesized successfully by the aid of NAR enzyme as a catalyzing and stabilizing agent in this bottom-up approach.”was revised to be clearer. The sentence was corrected to “In the present study, NAR enzyme was successfully utilized as a catalyzing and stabilizing agent to synthesize ZVI@graphene nanohybrid.”
- In figure 1, the sub-figures are not clearly generated, redraw the figures with high resolution for clarity.
Answer:
Thank you for your valuable comment. All figures were improved to be with high resolution.
- Same heading for section 2.4 and 3.2. It is really confusing.
Answer:
Thank you for your valuable comment. The heading in section 2.4 was changed to “Experimental procedures” to be different from the heading in section 3.2.
9.In section 3.5 Figure 5. There is no comparative inference on the effect of pH for three different sample studies. Improve the result analysis part.
Answer:
Thank you for your valuable comment. The discussion of results in section 3.5 was improved.
The following discussion was added:
“The effect of pH was studied by varying pH from 3 to 11 (Using 1 M of HCl or NaOH) at initial dye concentration of 10 mg/L, adsorbent dose of 40 mg/50 mL and shaking speed of 150 rpm as shown in Figure (5). Point of zero charge (PZC) of ZVI is nearly 8 as reported by Sun et al. (2015) [38]. At pH < PZC, the adsorbent’s surface charge becomes positive, and DR-81 is an ionic dye. Therefore, the removal efficiency was improved in acidic and neutral conditions due to the attractive forces between the adsorbent’s surface and DR-81 molecules. The elimination efficiency increased from 96.88±1.67% to 99.4±1.9% by decreasing pH from 7 to 3 using ZVI@graphene nanocomposite. On the other hand, the removal efficacy went down in alkaline conditions due to the repulsive forces between negatively charged adsorbent’s surface and anionic dye molecules. The removal efficiencies were 85.6±2.3% and 78.9±2.67% at pH 9 and 11, respectively using ZVI@graphene nanocomposite. In the case of MB, the removal efficiencies decreased at pH 3 and it was 81.4±1.9% due to the repulsive forces between positively charged sorbent’s surface and cationic dye. However, at neutral pH, the removal efficiency was high. The increase of pH from 3 to 7 resulted in the raising of hydroxyl ions which could be adsorbed on the adsorbent’s surface and consequently MB as a cationic dye could be adsorbed on the surface [39]. At high pH values, the removal efficiencies decreased to 88.7±1.4% and 80.1±1.1% at pH 9 and 11, respectively in spite of the attractive forces between the adsorbent and adsorbate. At alkaline conditions, ZVI could be easily corroded, and consequently iron hydroxides would be formed and precipitated [39]. The formed iron hydroxides could block the active sites and inhibit the electron transfer between ZVI and target pollutants which resulted in the decrease of removal efficiency [40]. Hamdy et al. (2018) reported the decrease of MB removal efficiency in alkaline condition due to the block of active sites of ZVI by the corrosion products [41]. In the case of the adsorption of MB and DR-81 over ZVI at pH > 7, the adsorption efficiency decreased due to the accelerated corrosion of ZVI resulting in the formation of iron hydroxides that could block the binding sites and suppress the electron transfer [40]. At pH ≤ 7, the adsorption ratio went up in the case of DR-81 due to the attraction between DR-81 molecules and ZVI surface, whereas the adsorption percentage declined in the case of MB due to the repulsive forces [39]. The adsorption at pH 3 was lower than that of pH 7 due to the competition between MB molecules and Cl- on ZVI surface [42]. At pH ≤ 7, the graphene surface charge is positive, whereas it was negative at pH > 7 as reported by Yang et al. (2022) [43]. So, the adsorption efficiency improved in the case of DR-81 at pH < 7 due to the attraction forces, while the adsorption ratio of DR-81 went down at pH > 7 due to the repulsive forces. In the case of MB, the adsorption proportion decreased at pH < 7 due to the repulsive forces and the adsorption went up at pH > 7 owing to the attractive forces. In the case of MB at pH 7, the adsorbed hydroxyl ions on the adsorbent’s surface might increase the attractive forces thus improving the degradation ratio.”
- Sections 3.4 and 3.6 are they come under the same methodology? Any differences between the two, and the discussion on results is not enough in both sections. These sections, 3.4-3.6 can be improved.
Answer:
Thank you for your valuable comment. The sections 3.4 and 3.6 were improvedto have different methodologies and the discussion was enhanced.
The discussion in “Effect of pH” section was improved as follows:
“The effect of pH was studied by varying pH from 3 to 11 (Using 1 M of HCl or NaOH) at initial dye concentration of 10 mg/L, adsorbent dose of 40 mg/50 mL and shaking speed of 150 rpm as shown in Figure (5). Point of zero charge (PZC) of ZVI is nearly 8 as reported by Sun et al. (2015) [38]. At pH < PZC, the adsorbent’s surface charge becomes positive, and DR-81 is an ionic dye. Therefore, the removal efficiency was improved in acidic and neutral conditions due to the attractive forces between the adsorbent’s surface and DR-81 molecules. The elimination efficiency increased from 96.88±1.67% to 99.4±1.9% by decreasing pH from 7 to 3 using ZVI@graphene nanocomposite. On the other hand, the removal efficacy went down in alkaline conditions due to the repulsive forces between negatively charged adsorbent’s surface and anionic dye molecules. The removal efficiencies were 85.6±2.3% and 78.9±2.67% at pH 9 and 11, respectively using ZVI@graphene nanocomposite. In the case of MB, the removal efficiencies decreased at pH 3 and it was 81.4±1.9% due to the repulsive forces between positively charged sorbent’s surface and cationic dye. However, at neutral pH, the removal efficiency was high. The increase of pH from 3 to 7 resulted in the raising of hydroxyl ions which could be adsorbed on the adsorbent’s surface and consequently MB as a cationic dye could be adsorbed on the surface [39]. At high pH values, the removal efficiencies decreased to 88.7±1.4% and 80.1±1.1% at pH 9 and 11, respectively in spite of the attractive forces between the adsorbent and adsorbate. At alkaline conditions, ZVI could be easily corroded, and consequently iron hydroxides would be formed and precipitated [39]. The formed iron hydroxides could block the active sites and inhibit the electron transfer between ZVI and target pollutants which resulted in the decrease of removal efficiency [40]. Hamdy et al. (2018) reported the decrease of MB removal efficiency in alkaline condition due to the block of active sites of ZVI by the corrosion products [41]. In the case of the adsorption of MB and DR-81 over ZVI at pH > 7, the adsorption efficiency decreased due to the accelerated corrosion of ZVI resulting in the formation of iron hydroxides that could block the binding sites and suppress the electron transfer [40]. At pH ≤ 7, the adsorption ratio went up in the case of DR-81 due to the attraction between DR-81 molecules and ZVI surface, whereas the adsorption percentage declined in the case of MB due to the repulsive forces [39]. The adsorption at pH 3 was lower than that of pH 7 due to the competition between MB molecules and Cl- on ZVI surface [42]. At pH ≤ 7, the graphene surface charge is positive, whereas it was negative at pH > 7 as reported by Yang et al. (2022) [43]. So, the adsorption efficiency improved in the case of DR-81 at pH < 7 due to the attraction forces, while the adsorption ratio of DR-81 went down at pH > 7 due to the repulsive forces. In the case of MB, the adsorption proportion decreased at pH < 7 due to the repulsive forces and the adsorption went up at pH > 7 owing to the attractive forces. In the case of MB at pH 7, the adsorbed hydroxyl ions on the adsorbent’s surface might increase the attractive forces thus improving the degradation ratio.”
Section 3.6 was revised and edited as follows:
“Figure 6 demonstrates the adsorption mechanism of DR-81 and MB on ZVI@graphene nanocomposite’s surface. The adsorption of MB on the adsorbent’s surface might be owing to the hydrogen bonding between OH and COOH groups of the adsorbent and NH group of MB [44]. In the case of DR-81, the adsorption took place via hydrogen bonding between SO3-, NH and N=N of DR-81 and hydroxyl group in the adsorbent [45]. Additionally, the hydrogen bonding could happen between COOH of the adsorbent and NH group of DR-81. The adsorption could take place due to the electrostatic interaction between COO- group (Deprotonated form of COOH) of the adsorbent and NH+ of MB. Moreover, the adsorption might occur due to the electrostatic interaction between Na+ and deprotonated form of COOH (COO-) in the case of DR-81 [46]. Further, the adsorption of MB or DR-81 might take place due to the π-π stacking between aromatic groups of MB or DR-81 and oxygen of hydroxyl group in the prepared adsorbent [45,47]. Additionally, the adsorption could take place due to the Yoshida hydrogen bonding between benzene ring of DR-81 or MB and H in hydroxyl group [45,48]. The decolorization of DR-81 and MB might also take place due to the reduction by atomic hydrogen (H*) formed owing to the reaction between electrons and H+ on the adsorbent’s surface [49]. Furthermore, the formed iron hydroxides could adsorb DR-81 and MB. The adsorption of DR-81 or MB on ZVI@grapheneis governed by hydrogen bonding, electrostatic interaction (attractive and repulsive forces), Yoshida hydrogen bonding and π-π stacking. So, the increase or the decrease of adsorption percentage can be affected by the attractive or repulsive forces between the adsorbent and adsorbate as explained in Section 3.5. The adsorption in the case of graphene could take place via the same mechanisms (hydrogen bonding, electrostatic interaction (attractive and repulsive forces), Yoshida hydrogen bonding and π-π stacking based on its functional groups as previously described in the literature [3].In the case of ZVI, adsorption could occur via hydrogen bonding, electrostatic interaction (attractive and repulsive forces), Yoshida hydrogen bonding and π-π stacking [49]. Additionally, the removal of dyes via ZVI might take place by reduction and precipitation [50].”
- The adsorption mechanism can be improved with the significant results obtained.
Answer:
Thank you for your valuable comment. The adsorption mechanism was improved.
The following discussion was added:
“Figure 6 demonstrates the adsorption mechanism of DR-81 and MB on ZVI@graphene nanocomposite’s surface. The adsorption of MB on the adsorbent’s surface might be owing to the hydrogen bonding between OH and COOH groups of the adsorbent and NH group of MB [44]. In the case of DR-81, the adsorption took place via hydrogen bonding between SO3-, NH and N=N of DR-81 and hydroxyl group in the adsorbent [45]. Additionally, the hydrogen bonding could happen between COOH of the adsorbent and NH group of DR-81. The adsorption could take place due to the electrostatic interaction between COO- group (Deprotonated form of COOH) of the adsorbent and NH+ of MB. Moreover, the adsorption might occur due to the electrostatic interaction between Na+ and deprotonated form of COOH (COO-) in the case of DR-81 [46]. Further, the adsorption of MB or DR-81 might take place due to the π-π stacking between aromatic groups of MB or DR-81 and oxygen of hydroxyl group in the prepared adsorbent [45,47]. Additionally, the adsorption could take place due to the Yoshida hydrogen bonding between benzene ring of DR-81 or MB and H in hydroxyl group [45,48]. The decolorization of DR-81 and MB might also take place due to the reduction by atomic hydrogen (H*) formed owing to the reaction between electrons and H+ on the adsorbent’s surface [49]. Furthermore, the formed iron hydroxides could adsorb DR-81 and MB. The adsorption of DR-81 or MB on ZVI@grapheneis governed by hydrogen bonding, electrostatic interaction (attractive and repulsive forces), Yoshida hydrogen bonding and π-π stacking. So, the increase or the decrease of adsorption percentage can be affected by the attractive or repulsive forces between the adsorbent and adsorbate as explained in Section 3.5. The adsorption in the case of graphene could take place via the same mechanisms (hydrogen bonding, electrostatic interaction (attractive and repulsive forces), Yoshida hydrogen bonding and π-π stacking based on its functional groups as previously described in the literature [3].In the case of ZVI, adsorption could occur via hydrogen bonding, electrostatic interaction (attractive and repulsive forces), Yoshida hydrogen bonding and π-π stacking [49]. Additionally, the removal of dyes via ZVI might take place by reduction and precipitation [50].”
Figure 6. Adsorption mechanism of DR-81 and MB over ZVI@graphene nanocomposite.
- All the sections need improvement with a clearer explanation.
Answer:
Thank you for your valuable comment. All sections were improved, and the discussion was supported with clearer explanation.
- Improve the resolution of the figures.
Answer:
Thank you for your valuable comment. The resolution of all figures was enhanced.
- The conclusion section can also be improved by using more significant findings of the study with futuristic applications of the study.
Answer:
Thank you for your valuable comment. The conclusion section was improved showing the main findings of the study and recommendations for future work.

Round 2
Reviewer 1 Report
1. The spelling and grammatical structures in the manuscript should be improved.
2. Based on my experiences, I suggest that the content of the abstract might be focused on the main ideas and shorter than it used to be. In brief, the abstract follows the format: (1) the purposes of the manuscript, (2) the main method used in this research, and (3) the principle results, major conclusion, and orient applied development. Thus, I suggest the authors make a revision to the abstract.
3. The format of the manuscript should be entirely at once. All paragraphs need to be formatted as justified alignment.
4. One of my biggest concerns about the study: is a lack of statistical analysis needs attention. It is necessary to add the statistical analysis for all results.
5. It is essential to provide a graphical abstract to summarize the whole research
6. Please give the formation mechanism of the ZVI@graphene nanocomposite using NAR enzymes
7. The TEM and SEM images of graphene precursors are required to be provided as controls with ZVI@graphene nanocomposite and given in the supplementary material file.
8. Also, the XRD and Raman patterns of graphene are added in the same diagrams of the ZVI@graphene nanocomposite for comparison.
9. The FITR spectroscopy and element mapping of graphene and the ZVI@graphene nanocomposite are carried out to determine the composition and the elemental distribution in the samples, especially iron.
10. I refer to the specific surface area (BET) calculation of the ZVI@graphene nanocomposite that must be added.
11. I worry about that how can the authors demonstrate the stability of the ZVI@graphene nanocomposite.
12. The authors must be provided in detail the kinetics and equilibrium of the other adsorption models instead of the pseudo-first-order model.
13. After the adsorption process, did the desorption process be conducted?
14. The authors need to provide SEM and FESEM images, FTIR spectra, XRD patterns, Raman spectra, and element mapping of the ZVI@graphene nanocomposite after both adsorption and desorption processes with respect to DR-81 and MB. The explanation of these results is also analyzed as required.
15. I suggest that the authors should be constructed the structure and rearranged the results, if possible, for logic and rationality throughout the manuscript. It can be available as a supplementary material file in addition to containing the supporting results.
16. I recommend the authors refer to the following studies for a clearer and more detailed explanation of the DR-81 and MB adsorption mechanism of the ZVI@graphene nanocomposite. Individuals the adsorption mechanism of graphene, zero-valent iron is required. Then, the generally simultaneous mechanism of both components should be proposed as the provable feasibility of the mechanism: https://doi.org/10.1016/j.surfin.2021.101023, https://doi.org/10.3390/ijerph17165817, https://doi.org/10.3390/nano10050917, https://doi.org/10.1016/j.enmm.2021.100584.
Author Response
We have already responded at the first round, please see the attachment.

Reviewer 2 Report
Manuscript Number: Sustainability-1908385-v3
Title: “Novel biosynthesis of graphene-supported zero-valent iron nanohybrid for efficient decolorization of acid and basic dyes” by Mahmoud Samy et al.
The authors have done significant changes by considering the reviewer’s suggestion. I have gone through every section of the manuscript, and now the manuscript has reached its best possible level after significant improvements.
Please check the journal format and must do the final proofreading before accepting the manuscript for publication.
And take care of the figure's resolution quality.
With the above comments, I recommend acceptance of the manuscript.
Author Response
Dear
Editor-in-Chief
Sustainability - MDPI
We are pleased to send our responseto the comments of reviewers, included at the bottom of this letter.All added text and modifications are highlighted in the revised manuscript. We have answered point-by-point response to these comments as follows,
_____________________________________________________________
Reviewer #2:
Comment:
The authors have done significant changes by considering the reviewer’s suggestion. I have gone through every section of the manuscript, and now the manuscript has reached its best possible level after significant improvements.
Answer:
Thank you for your response.
- Please check the journal format and must do the final proofreading before accepting the manuscript for publication.
Answer:
Thank you for your valuable comment.We revised the manuscript to agree with Sustainability format. Additionally, we checked the manuscript carefully to correct any grammatical errors.
- And take care of the figure's resolution quality.
Answer:
Thank you for your valuable comment. The resolution of all figures was improved.
Round 3
Reviewer 1 Report
I agree and accept publication of the manuscript without any revision
Author Response
Dear
Editor-in-Chief
Sustainability - MDPI
We are pleased to send our responseto the comments of reviewers, included at the bottom of this letter.All added text and modifications are highlighted in the revised manuscript. We have answered point-by-point response to these comments as follows,
_____________________________________________________________
Reviewer #1:
Comment:
I agree and accept publication of the manuscript without any revision
Thank you for your response.